# First-Line Use of Daratumumab in Patients with Multiple Myeloma Shows Delayed Neutrophil and Platelet Engraftment after Autologous Stem Cell Transplantation: Results from a Real-Life Single-Center Study

**DOI:** 10.3390/cancers16193307

**Published:** 2024-09-27

**Authors:** Massimo Martino, Mercedes Gori, Gaetana Porto, Giorgia Policastro, Martina Pitea, Annalisa Sgarlata, Ilaria Maria Delfino, Francesca Cogliandro, Anna Scopelliti, Giovanna Utano, Maria Pellicano, Aurora Idato, Iolanda Donatella Vincelli, Violetta Marafioti, Maria Caterina Micò, Giuseppe Lazzaro, Barbara Loteta, Caterina Alati, Giovanni Leanza, Graziella D’Arrigo, Giovanni Luigi Tripepi, Annalisa Pitino

**Affiliations:** 1Hematology and Stem Cell Transplantation and Cellular Therapies Unit (CTMO), Department of Hemato-Oncology and Radiotherapy, Grande OspedaleMetropolitano“Bianchi-Melacrino-Morelli”, 89133 Reggio Calabria, Italy; massimo.martino@ospedalerc.it (M.M.);; 2Stem Cell Transplant Program CIC587, 89124 Reggio Calabria, Italy; 3Institute of Clinical Physiology (IFC-CNR), Section of Rome, 00185 Rome, Italyannalisa.pitino@cnr.it (A.P.); 4Pharmacy Unit, Grande Ospedale Metropolitano “Bianchi-Melacrino-Morelli”, 89133 Reggio Calabria, Italy; 5Institute of Clinical Physiology (IFC-CNR), Section of Reggio Calabria, 89124 Reggio Calabria, Italy

**Keywords:** multiple myeloma, daratumumab, autologous stem cell transplantation, neutrophil and platelet engraftment

## Abstract

**Simple Summary:**

Daratumumab (DARA) plus bortezomib, thalidomide, and dexamethasone (D-VTd) represent the standard of induction care in Europe for autologous stem cell transplantation (auto-SCT)-eligible, newly diagnosed multiple myeloma (NDMM) patients. This study aimed to investigate the possible impact of D-VTd induction therapy on hematopoietic engraftment after auto-SCT. Our findings indicate that patients treated with D-VTd experienced longer neutrophil and platelet engraftment times than those treated with VTd. Additionally, D-VTd treatment was associated with a higher incidence of febrile neutropenia and grade 2 or higher diarrhea. However, no significant differences were observed in the median number of days to discharge. The conclusion we can draw from our real-life study is that a four-drug induction therapy containing DARA does not impact transplant safety outcomes.

**Abstract:**

Background: This real-life study aimed to investigate the possible impact of D-VTd induction therapy on hematopoietic engraftment after autologous stem cell transplantation (auto-SCT). Methods: Sixty consecutive NDMM patients received four cycles of induction therapy with D-VTd. The conditioning regimen consisted of melphalan 200 mg/m^2^. These patients were compared with a historical control group of 80 patients who received four cycles of VTd as induction therapy. Results: The median days to reach neutrophil and platelet engraftment significantly differed between patients treated with D-VTd (11 and 13 days, respectively) and VTd (10 and 12 days). Univariate Cox analyses show that patients treated with D-VTd had a hazard ratio of neutrophil engraftment that was 42% significantly lower than those in the VTd arm (HR: 0.58, *p* = 0.002), and a multivariate model confirmed this result. Patients treated with D-VTd developed FN more frequently. Univariate and multivariate logistic regressions revealed an association between D-VTd and FN. Delayed engraftment did not correlate with more extended hospitalization. No patients died in the first six months after transplantation. Conclusions: Our real-life study showed that a four-drug induction therapy containing DARA does not impact transplant safety outcomes.

## 1. Introduction

Multiple myeloma (MM) is the second most common hematologic malignancy worldwide, with an estimated annual incidence of 7.1 per 100,000 people [1]. Treatment options have expanded widely over the past several years, significantly improving outcomes [2]. For newly diagnosed (ND) eligible patients, the three-drug induction regimens with immunomodulatory (IMiDs), proteasome inhibitors, and dexamethasone followed by autologous stem cell transplantation (auto-SCT) and maintenance treatment have shown significantly improved progression-free survival (PFS) and overall survival (OS) [3]. Two decades ago, the median survival was approximately three years, and now it is eight to ten years and can be even longer for many patients.

The advent of four-drug regimens has raised the question of whether certain patients would benefit from the addition of another drug. Daratumumab (DARA) is a human IgG monoclonal antibody targeting CD38 on clonal plasma cells with direct on-tumor and immunomodulatory mechanisms of action [4]. DARA-based combination induction therapy’s clinical efficacy and safety for transplant-eligible NDMM patients was first investigated in the phase II Griffin trial [5]. The addition of DARA to lenalidomide (R), bortezomib (V), and dexamethasone (d) (D-RVd) improved the depth of response and PFS in this setting of patients. DARA plus V, thalidomide, and d (D-VTd) were validated in the phase III CASSIOPEIA trial, which compared VTd for induction and post-transplant consolidation without or with the anti-CD38 antibody DARA [6,7]. The stringent complete response (sCR) rates, measurable residual disease (MRD) negativity, and PFS improved when DARA was added to pre-transplant induction and post-transplant consolidation therapy. Currently, D-VTd represents the standard of care in Europe for transplant-eligible NDMM patients. DARA could be involved in CD38 expression on CD34+ cells, possibly to affect mobilization kinetics and lineage-specific progenitor proliferative capacity, and recent studies have reported a potential reduction in stem cell yields in patients who were exposed to DARA before stem cell mobilization [8,9]. This regimen also increased the use of the hematopoietic stem cell mobilizer plerixafor in this population [10,11]. Moreover, some recently published studies have reported slower hematopoietic reconstitution after auto-SCT in patients treated with DARA, without an excess of infectious complications, with the limitation that the criteria for defining platelet and neutrophil engraftment were not uniform [8,10,12,13,14,15,16,17,18,19]. This real-life single-center study aimed to investigate the possible impact of D-VTd induction therapy on hematopoietic engraftment after auto-SCT.

## 2. Materials and Methods

This study was conducted at the the Stem Cell Transplantation Unit of the Grande Ospedale Metropolitano “Bianchi-Melacrino-Morelli” (GOM-BMM) in Reggio Calabria, Italy, as an observational retrospective analysis of drug effectiveness and safety, and in accordance with applicable guidelines, ethical committee approval was not deemed necessary. However, informed consent was obtained from all participants to ensure ethical standards were maintained.

### 2.1. Patients

This real-world study compared patients receiving D-VTd induction with a group of historical controls, all of whom were referred to the GOM-BMM for peripheral blood stem cell collection and auto-SCT. The study included NDMM patients eligible for a transplantation procedure who were aged 18–70 years and who achieved a favorable response after induction therapy. Complete response (CR), very good partial response (VGPR), and partial response (PR) were defined according to the International Myeloma Working Group criteria [20]. Patients were excluded if they met any of the following criteria: a World Health Organization performance status of >2; non-secretory MM; Waldenstrom macroglobulinemia or immunoglobulin M MM; New York Heart Association class II–IV heart failure; abnormal pulmonary function findings; systematic amyloid light-chain amyloidosis; or a history of active malignancy during the past five years (excluding basal cell carcinoma or stage 0 cervical cancer). Patients with an absolute neutrophil count (ANC) of ≤1.0 × 10^9^/L, a platelet count of ≤75 × 10^9^/L, abnormal renal function (serum creatinine value) > 2.0 mg/dL, and those with disease refractory to induction chemotherapy were also excluded.

### 2.2. Treatment

After diagnosis, 60 consecutive NDMM patients were treated with four cycles of D-VTd according to the CASSIOPEIA trial schedule [6,7] before stem cell mobilization and transplantation. All patients received up to four 28-day, pre-transplant induction cycles of: V (1·3 mg/m^2^ twice per week in week 1 (days 1 and 4) and week 2 (days 8 and 11) of each cycle) administrated by subcutaneous bolus injection into the thigh or abdomen; oral T (100 mg daily in all cycles); and oral or intravenous d (40 mg on days 1, 2, 8, 9, 15, 16, 22, and 23 of induction cycles 1 and 2 and days 1 and 2 of induction cycles 3 and 4 and 20 mg on days 8, 9, 15, and 16 of induction cycles 3 and 4). D was administered at a dose of 1800 mg SC bolus once weekly in induction cycles 1 and 2 and once every 2 weeks during induction cycles 3 and 4. DARA was injected into the subcutaneous tissue of the abdomen approximately 7.5 cm to the right or left of the navel over approximately 3–5 min. The study population has been compared with a historical group of 80 NDMM treated with VTd as induction therapy, who were treated between February 2021 to December 2023 without the use of DARA [6,7]. High-dose cyclophosphamide (2 g/m^2^) and G-CSF were administered to mobilize peripheral blood stem cells in both arms. The conditioning regimen consisted of melphalan (MEL) 200 mg/2 for all patients. The minimum target dose of CD34+ cells required to support MEL safely was 2 × 10^6^/kg. Patients received a single 6 mg subcutaneous injection of BIO/PEG 24 h after stem cell infusion. These patients were compared with a historical control group of patients who received four cycles of VTd as induction therapy from January 2019 to January 2021. All patients received oral prophylaxis with levofloxacin 500 mg/day from day 0 until neutrophil engraftment and acyclovir 800 mg twice daily from day 3 until approximately day 90 post-transplantation. Pneumocystis jirovecii pneumonia prophylaxis was administered with trimethoprim/sulfamethoxazole (1 double-strength tablet; 2–3 times weekly) and initiated post-hematologic recovery for three months. Cryotherapy (ice chips) was utilized to prevent MEL-induced oral mucositis; the patients placed ice chips in their mouths approximately 30 min before and 6 h after MEL. Red blood cell (RBC) and platelet transfusions (PT) were administered to maintain hemoglobin levels of ≥8 mg/dL and platelet counts of ≥10 × 10^9^/L or in patients with symptomatic anemia/minimal mucocutaneous hemorrhagic syndrome. Intravenous hydration and electrolyte support were also provided. Where febrile neutropenia (FN) occurred following a long period of neutropenia (ANC < 0.5 × 10^9^/L or ANC of 1 × 10^9^/L with a predicted decline to <0.5 × 10^9^/L over the subsequent 48 h), blood and catheter-drawn cultures were ordered, and intravenous ceftriaxone was promptly started.

### 2.3. Endpoints

The primary endpoints of this study were time to hematological recovery, including neutrophil recovery, defined as the first of three consecutive days with an absolute neutrophil count ≥0.5 × 10^9^/L, and platelet recovery, described as a platelet count of ≥20 × 109/L in the absence of PT for seven consecutive days [21]. Complete blood counts were collected using samples before chemotherapy and daily during the aplastic phase until hospital discharge. Secondary endpoints included the incidence and duration of FN, the incidence of mucositis and diarrhea, and the duration of hospitalization. FN was defined as a temperature of ≥38.2 °C on at least two consecutive occasions or a persistent temperature of ≥38.0 °C for at least one hour, accompanied by an ANC of <0.5 × 10^9^/L in the absence of any documented noninfectious cause (e.g., transfusion reaction or administration of cytotoxic drugs) [22]. In the absence of clinically or microbiologically documented infection, empirical antibiotic was discontinued after 72 h of apyrexia and clinical recovery, irrespective of the neutrophil count. If no signs or symptoms of clinical deterioration were present, slow response to antibiotic treatment was considered, particularly if accompanied by improvements in inflammatory markers such as C-reactive protein or procalcitonin. If clinical conditions deteriorated, management steps included an aggressive diagnostic workup (repeated blood cultures, additional testing for viruses and fungi, CT scan, BAL lavage in case of pneumonia, and lumbar puncture in case of CNS symptoms).

### 2.4. Statistical Analysis

Data were expressed as mean and standard deviation (SD), median and interquartile range (IQR), and percentages; comparisons between groups were performed using the Mann–Whitney test or chi-squared test, as appropriate (see Table 1 and Table 2 and Appendix A). Covariates to be introduced into multivariable models and associated with treatment (DARA yes/no), FN (yes/no), diarrhea (WHO grade <1 or >2), or mucositis (WHO grade <1 or >2) were identified by comparative analyses. Multivariable logistic regression analyses were conducted to assess the simultaneous effects of variables significantly associated with treatment, including age and gender. Data were expressed as odds ratios (ORs), 95% CI, and p values. To assess the relationship between therapy (D-VTd vs. VTd), time to neutrophil and platelet engraftment, and time to discharge, both Kaplan–Meier analysis and univariate and multivariate (age and gender-adjusted) Cox models were utilized. In the Cox models, data were expressed as hazard ratios (HRs), 95% CI, and p values. Statistical analyses were performed using the R Survival package, version 4.2.3, and SPSS 29.

## 3. Results

The study population included 140 patients. Among these, 60 (43%) were treated with D-VTd (Table 1). Patients treated with D-VTd significantly differed from those treated with VTd in the number of basal CD34+ infused, either as a continuous variable or as categorized in binary terms (<4 and ≥4 × 10^6^/kg). Indeed, the median number of basal CD34+ infused was significantly lower (and the proportion of patients with CD34 <4 × 10^6^/kg was higher) in patients treated with D-VTd than those treated with VTd. No differences were found for the remaining variables listed in Table 1, including disease status.

### 3.1. Analysis of Outcome Variables by Treatment

The median days to reach neutrophil and platelet engraftment significantly differed between patients in the D-VTd arm (11 and 13 days, respectively) compared with the VTd arm (10 and 12 days, respectively) (Table 2). In Figure 1, the number of patients reaching neutrophil engraftment is plotted as a function of days to neutrophil engraftment (ANC ≥ 0.5 × 10^9^/L) in the overall group and separately in D-VTd and VTd groups. The highest number of patients achieving neutrophil engraftment was observed on day 11 (*n* = 20) in the D-VTd arm and on day 10 (*n* = 28) in the VTd arm.

Remarkably, univariable Cox analyses (Table 3a) show that patients treated with D-VTd had a hazard ratio of neutrophil engraftment that was 42% lower than those treated with VTd (HR: 0.58, 95% CI: 0.41–0.82, *p* = 0.002), and these results did not change in multivariable age- and sex-adjusted analysis (Table 3b). Univariable Cox analyses (Table 4a) indicate that patients receiving D-VTd treatment and those with micromolecular characteristics displayed significantly longer times to platelet engraftment (HR: 0.62, 95% CI: 0.43–0.89 for therapy, HR 0.46, 95% CI: 0.25–0.83 for myeloma type). Multivariable analysis fully confirmed these results (Table 4b). The hazard to engraftment for patients affected by the micromolecular type was 57% lower than those with the IgG subtype (HR 0.43, 95% CI: 0.23–0.79) and 37% lower in patients treated with D-VTd compared to those who received VTd (HR 0.63, 95% CI: 0.44–0.92).

### 3.2. FN, Mucositis, and Diarrhea

Patients treated with D-VTd developed FN more frequently than those in the VTd group. Notably, patients on D-VTd had a lower incidence of a WHO fever > 2 and a higher incidence of WHO diarrhea> 2. The incidence of mucositis tended to be higher in the D-VTd arm (*p* = 0.094). Remarkably, patients with FN more frequently had WHO diarrhea > 2 grade (Table 5). Univariable and multivariable logistic regression analyses confirmed the strong association between D-VTd and FN (Table 6a and b, respectively). Specifically, the likelihood of developing febrile neutropenia in patients receiving D-VTd was more than twice as high as those treated with VTd (OR 2.24, 95% CI: 1.12–4.47).

A higher WHO diarrhea grade was associated with D-VTd treatment (Table 2), but unrelated to the remaining baseline characteristics. Univariable and multivariable logistic models confirmed the link between WHO diarrhea grade and D-VTd treatment. Patients receiving D-VTd were eight times more likely to develop grade 2 or higher diarrhea compared to those who did not receive treatment (Table 7).

No associations were found between D-VTd treatment, mucositis, and the other baseline characteristics.

### 3.3. Transfusions

The proportion of patients who underwent RBC did not differ between the treatment groups, while patients treated with D-VTd underwent PT more frequently than those treated with VTd. There was no difference in the median number of platelet-transfused patients (Table 2).

### 3.4. Discharge

The median number of days to discharge was unaffected by treatment (13 days for both groups; Table 2), and Kaplan–Meier analysis of time to discharge confirmed this result (median: 13 days, 95% CI: 12.5–13.5 in VTd group; median: 13 days, 95% CI: 12.4–13.6, in D-VTd group) (Log-Rank test *p* = 0.99). No patients experienced unexpected side effects during transplantation. In particular, no cardiac, renal, or hepatic toxicities were reported.

During the first 100 days after transplantation, we observed four COVID-19 infections, two in both arms. Patients developed a symptomatology with fever and cough and without respiratory failure, and were treated with specific symptomatic and antiviral drugs, with rapid resolution of the clinical picture. Between 100 and 180 days post-transplant, three patients developed radiologically proven pneumonia with FUO, two in the D-VTd arm. Patients were managed as outpatients with rapid resolution of the picture.No patients died during the first six months of follow-up.

## 4. Discussion

Eliminating auto-SCT is a topic of discussion among opinion leaders in MM. However, it is not currently advised, and transplant remains a standard of care for eligible MM patients, as recommended by the most recent guidelines [23]. In 2022, 27,132 auto-SCTs were reported by 689 European centers [24]. The main indications for auto-SCT were lymphoid malignancies, with MM comprising 57.1% of all auto-SCT indications. Auto-SCT activities for lymphoproliferative disorders increased by +2.4% for MM (+4.8% in 2021) and declined for non-Hodgkin lymphoma by −10.5% (+4.3% in 2021).

Auto-SCT in MM continues to survive and thrive, even in the context of new treatments. This is because every trial comparing transplant to no transplant, even with new drugs, has shown that transplant deepens the response and offers significant benefits [25,26,27,28,29]. Auto-SCT was also considered in the trials of DARA-based quadruplets as induction therapy in MM [5,6,7] and in studies investigating other anti-CD38 MoAb-based therapies in the same setting [30,31,32].

While no direct effect on stem cells was observed in vitro, emerging evidence suggests possible dysregulation of CD34+ cell adhesion after DARA treatment. Overall, anti-CD38 monoclonal antibodies appear to interfere with CD34+ cell mobilization, with no apparent clinical consequences during the transplantation phase.

A comparison between the various studies in terms of post-transplant engraftment is difficult because of the different definitions used for platelet and neutrophil recovery. In several studies, platelet engraftment was significantly slower in the DARA group, while no significant differences were reported in other trials [12,13,14,15,16,17]. Similarly, neutrophil recovery was significantly slower in patients treated with DARA in different studies [8,10,18,19,20], but not significantly different in others [13,18,33]. The delay in hematopoietic engraftment was typically 1 or 2 days, but all patients achieved hematopoietic recovery. Regarding transfusion requirements, study results are conflicting, showing in some cases the need for increased PT in those who had received DARA [15]. Additionally, in some trials, DARA-treated patients received more RBC transfusions [24], while in other studies, transfusion rates were similar between the DARA and control groups [13,17].

The rates of neutropenic fever were comparable between DARA and control patients across all studies. No significant differences in severe infections, antibiotic therapy duration, or hospitalization length were observed [9,16,17,19,33]. However, Papaiakovou et al. reported longer durations and the need for more lines of antibiotic therapy, higher incidence of septic shock, and prolonged hospitalization in the patients treated with DARA [15]. This did not translate into higher transplant-related mortality rates. The authors suggested that this excess risk was unlikely to be solely explained by the slight delay in neutrophil recovery, and they hypothesized that DARA might worsen immunosuppression through hypogammaglobulinemia and lymphodepletion.

Our findings indicate that patients treated with D-VTd experienced longer neutrophil and platelet engraftment times than those treated with VTd. Additionally, D-VTd treatment was associated with a higher incidence of febrile neutropenia and grade 2 or higher diarrhea. However, no significant differences were observed in the median number of days to discharge or the incidence of mucositis between the two treatment groups. Despite the study’s limitations and observational design, these results provide valuable insights into the differential effects of D-VTd and VTd treatments. It is important to note that, although the median number of CD34+ cells infused was significantly lower in the D-VTd group compared to the VTd group, the magnitude of this difference was very small from a clinical perspective (D-VTd: 4.6 versus VTd: 4.9). Therefore, it does not explain the differential outcomes observed between patients treated with D-VTd and those treated with VTd. In any case, the number of CD34+ cells infused was considered in all multivariable models as a potential confounder, thus excluding the possibility that this variable could influence the study results.

Although not a goal of this study, we evaluated infectious events that occurred after engraftment for neutrophils and within six months post-transplantation, showing no difference in incidence between the two study groups. We follow the current indications of the European Society for Blood and Marrow Transplantation (EBMT), which recommend revaccination starting between 6 and 12 months after transplant [34]. To reduce the risk of SARS-CoV-2 infection, the primary immunization schedule consisted of three vaccine doses starting from 3 to 6 months after transplant, followed by a booster dose after 3–4 months from the primary vaccine schedule.

Our study was not aimed at assessing survival, so we did not consider data related to patients’ cytogenetic risk in the statistical analysis. With more mature follow-up, it will be interesting to consider data in terms of PFS and OS, stratifying patients according to cytogenetic features at disease onset.

## 5. Conclusions

The conclusion we can draw from our real-life study is that a four-drug induction therapy containing DARA does not impact transplant safety outcomes. No patients died in the first six months after transplantation, and we did not observe a higher incidence of infection disease in the D-VTd arm. Specifically, considering that MM has been the prototypical disease in which auto-SCT has been performed following outpatient approaches [35,36,37], D-VTd therapy does not preclude the continuation of this pathway. Further randomized controlled trials with larger sample sizes and more extended follow-up periods are needed to confirm these findings and better understand the long-term impact of these treatments on patient outcomes.

## Figures and Tables

**Figure 1 cancers-16-03307-f001:**
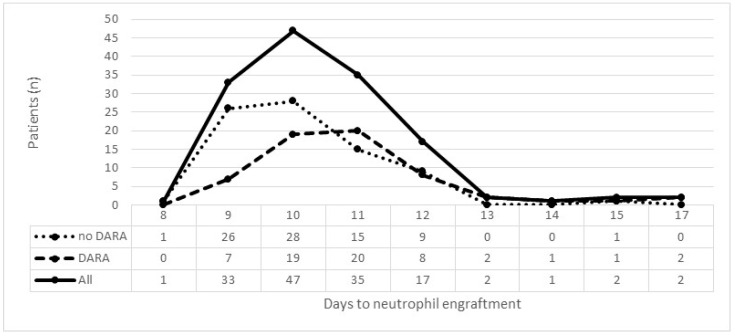
Absolute number of patients reaching neutrophil engraftment (absolute neutrophil count ≥0.5 × 10^9^/L) by treatment group.

**Table 1 cancers-16-03307-t001:** Baseline characteristics by treatment.

	All (*n* 140)	Vtd (*n* 80)	D-VTd (*n* 60)	*p* Value
Gender (male), *n* (%)	75 (53.6)	42 (52.5)	33 (55)	0.769
Age at transplant, mean (SD)	58.5 (7.7)	59 (7.9)	57.8 (7.5)	0.254
Myeloma type				
IgG, *n* (%)	98 (70)	52 (65)	46 (76.7)	0.322
IgA, *n* (%)	26 (18.6)	17 (21.3)	9 (15)	
Micromolecular, *n* (%)	16 (11.4)	11 (13.8)	5 (8.3)	0.424
Disease status at transplant				
CR/VGPR, *n* (%)	51 (83.57)	28 (83.75)	23 (83.33)	0.95
PR, *n* (%)	23 (16.43)	13 (16.25)	10 (16.67)	
ISS classification				
I stage, *n* (%)	33 (23.6)	18 (22.5)	15 (25)	0.63
II stage, *n* (%)	74 (52.9)	45 (56.3)	29 (48.3)	
III stage, *n* (%)	33 (23.6)	17 (21.3)	16 (26.7)	
No. of PLT, median (IQR)	189,000 (156,000–222,000)	189,000 (156,000– 221,000)	189,000 (155,000–230,000)	0.98
No. of WBC, median (IQR)	5200 (4500–6000)	5100 (4500–6000)	5300 (4500–6000)	0.99
CD34+ infused				
Median (IQR)	4.7 (3.6–5.6)	4.9 (4–6)	4.6 (3.3–5.3)	0.008
CD34 < 4, *n* (%)	43 (30.7)	19 (23.8)	24 (40)	0.039

**Legend**: Daratumumab (D); Velcade (V), thalidomide (T); dexamethasone (d); complete remission (CR); very good partial remission (VGPR); partial remission (PR); International Staging System (ISS).

**Table 2 cancers-16-03307-t002:** Outcome measurements according to treatments.

	All	VTd	D-VTd	*p* Value
Febrile neutropenia, no. patients (%)	64 (45.7)	30 (37.5)	34 (56.7)	0.024
Fever, WHO grade				
1, *n* (%)	47 (73.4)	18 (60)	29 (85.3)	0.022
≥2, *n* (%)	17 (26.6)	12 (40)	5 (14.7)	
Cause of fever				
FUO, *n* (%)	43 (67.2)	18 (60)	25 (73.5)	0.25
Documented, *n* (%)	21 (32.8)	12 (40)	9 (26.5)	
Median (IQR), days of fever	2 (1–2.5)	2 (1–2)	2 (1–3)	0.972
Mucositis, WHO grade				
0–1, *n* (%)	120 (85.7)	72 (90)	48 (80)	0.094
≥2, *n* (%)	20 (14.3)	8 (10)	12 (20)	
Diarrhea, WHO grade				
0–1, *n* (%)	118 (84.3)	76 (95)	42 (70)	<0.001
≥2, *n* (%)	22 (15.7)	4 (5)	18 (30)	
Nausea, WHO grade				
0, *n* (%)	100 (71.4)	59 (73.8)	41 (68.3)	0.349
1, *n* (%)	29 (20.7)	17 (21.3)	12 (20)	
≥2, *n* (%)	11 (7.9)	4 (5)	7 (11.7)	
Vomiting, WHO grade				
0, *n* (%)	136 (97.1)	80 (100)	56 (93.3)	0.019
≥1, *n* (%)	4 (2.9)	0 (0)	4 (6.7)	
Patients who required RBC transfusion	26 (18.6)	14 (17.5)	12 (20)	0.707
No. of RBC transfusions, median (IQR)	1 (1–2)	1.5 (1–2)	1 (1–3)	0.899
Patients who required PLT transfusion, *n* (%)	69 (49.3)	33 (41.3)	36 (60)	0.028
No. of PLT transfusions, median (IQR)	1 (1–1)	1 (1–1)	1 (1–2)	0.129
Median (IQR) days to neutrophil engraftment (ANC ≥ 0.5 × 10^9^/L)	10 (10–11)	10 (9–11)	11 (10–11)	<0.001
Median (IQR) days to reach PLT count ≥20 × 10^9^/L	12 (11–14)	12 (11–14)	13 (11.5–14)	0.02
Median (IQR) days to discharge	13 (12–14)	13 (12–14)	13 (12–15)	0.236
Median (IQR) days with neutrophil <100	3 (2–4)	3 (2–3)	3 (2–4)	0.017
Median (IQR) days with neutrophil <500	4 (3–5)	4 (3–5)	5 (3.5–5.5)	0.001
Median (IQR) days with neutrophil <1000	5 (4–6)	4 (4–5)	6 (5–7)	<0.001

**Legend:** fever unknown origin (FUO); red blood cells (RBC); platelet (PLT).

**Table 3 cancers-16-03307-t003:** Cox analyses of time to neutrophil engraftment.

**(a) Univariable Cox Analysis.**		
	**HR (95% CI)**	** *p* **
Gender (M vs. F)	1.09 (0.78–1.52)	0.617
Age at transplant	1.02 (1–1.04)	0.110
Disease status at transplant PR vs. CR/VGPR	0.67 (0.43–1.06)	0.088
CD34+ ≥4 vs. <4	1.40 (0.97–2.01)	0.071
D-VTd vs. VTD	0.58 (0.41–0.82)	0.002
**(b) Multivariable Cox Analysis.**		
	**HR (95% CI)**	** *p* **
Gender (M vs. F)	1.02 (0.72–1.43)	0.928
Age at transplant	1.01 (0.99–1.04)	0.299
Disease status at transplant PR vs. CR/VGPR	0.74 (0.45–1.19)	0.215
CD34+ ≥4 vs. <4	1.17 (0.80–1.71)	0.420
D-VTd vs. VTD	0.59 (0.41–0.84)	0.003

**Legend**: Complete remission (CR); very good partial remission (VGPR); partial remission (PR); International Staging System (ISS).

**Table 4 cancers-16-03307-t004:** Cox analyses of time to platelet engraftment.

**(a) Univariable Cox Analysis.**		
	**HR (95% CI)**	** *p* **
Gender (male vs. female)	1.03 (0.73–1.46)	0.871
Age at transplant	1 (0.98–1.03)	0.902
Myeloma IgA vs. IgG	0.96 (0.62–1.5)	0.859
Myeloma micromolecular vs. IgG	0.46 (0.25–0.83)	0.010
Disease status at transplant PR vs. CR/VGPR	1.11 (0.68–1.78)	0.683
CD34+ ≥4 vs. <4	1.54 (1.04–2.27)	0.031
D-VTd versus VTd	0.62 (0.43–0.89)	0.010
**(b) Multivariable Cox Analysis.**		
	**HR (95% CI)**	** *p* **
Gender (M vs. F)	0.89 (0.61–1.28)	0.522
Age at transplant	0.99 (0.97–1.02)	0.530
Myeloma IgA vs. IgG	0.92 (0.58–1.45)	0.721
Micromolecular vs. IgG	0.43 (0.23–0.79)	0.007
CD34+ ≥4 vs. <4	1.36 (0.89–2.06)	0.153
D-VTd versus VTd	0.63 (0.44–0.92)	0.017

**Legend**: Complete remission (CR); very good partial remission (VGPR); partial remission (PR).

**Table 5 cancers-16-03307-t005:** Main clinical characteristics according to neutropenia febrile.

	All	No FN	FN	*p* Value
Gender male *n* (%)	75 (53.6)	39 (51.3)	36 (56.3)	0.56
Age at transplant mean (SD)	58.5 (7.7)	58.1 (7.5)	59 (8.1)	0.297
CD34+ < 4	43 (30.7)	22 (28.9)	21 (32.8)	0.621
Mucositis ≥ 2 grade	20 (14.3)	9 (11.8)	11 (17.2)	0.368
Diarrhea ≥ 2 grade *n* (%)	22 (15.7)	5 (6.6)	17 (26.6)	0.001
Nausea 0 grade *n* (%)	100 (71.4)	53 (69.7)	47 (73.4)	0.793
Nausea 1 grade *n* (%)	29 (20.7)	16 (21.1)	13 (20.3)	
Nausea ≥ 2 grade *n* (%)	11 (7.9)	7 (9.2)	4 (6.3)	

**Table 6 cancers-16-03307-t006:** Logistic regression models of FN.

**(a) Univariable Logistic Analysis.**		
	Univariate	
	OR (95% CI)	*p* value
Gender (M vs. F)	1.22 (0.63–2.38)	0.56
Age at transplant	1.02 (0.97–1.06)	0.47
CD34+ ≥4 vs. <4	0.83 (0.41–1.71)	0.62
D-VTd vs. VTd	2.18 (1.1–4.31)	0.03
**(b) Multivariable Logistic Analysis.**		
	Multivariable	
	OR (95% CI)	*p* value
Gender (male vs. female)	1.19 (0.6–2.35)	0.62
Age at transplant	1.02 (0.98–1.07)	0.37
D-VTd vs. VTd	2.24 (1.12–4.47)	0.02

**Table 7 cancers-16-03307-t007:** Univariable and multivariable logistic regression models of diarrhea WHO ≥2 vs. <2 grade.

**(a) Univariable Logistic Analysis**		
	Univariable	
	OR (95% CI)	*p* value
Gender (male vs. female)	0.68 (0.27–1.7)	0.407
Age at transplant	1.01 (0.95–1.08)	0.691
D-VTd vs. VTd	8.14 (2.59–25.64)	0.000
CD34+ ≥4 vs. <4	0.47 (0.18–1.18)	0.108
**(b) Multivariable Logistic Analysis.**		
	Multivariable	
	OR (95% CI)	*p* value
Gender (male vs. female)	0.59 (0.22–1.59)	0.300
Age at transplant	1.03 (0.96–1.1)	0.399
D-VTd vs. VTd	8.81 (2.75–28.25)	0.000

## Data Availability

Raw data were generated at the Institute of Clinical Physiology (IFC-CNR), Reggio Calabria, Italy. Derived data supporting the findings of this study are available from the corresponding author upon request.

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
