# Peer review of "First-Line Use of Daratumumab in Patients with Multiple Myeloma Shows Delayed Neutrophil and Platelet Engraftment after Autologous Stem Cell Transplantation: Results from a Real-Life Single-Center Study"

_cancers, 2024, doi:10.3390/cancers16193307_

Round 1

Reviewer 1 Report

Comments and Suggestions for Authors

This manuscript by Martino et al. is of valuable clinical significance as it adds more data regarding the safety and feasibility of Daratumumab-implemented immunochemotherapy of ASCT-eligible patients. The authors confirmed a previously-reported slight delay in platelet and neutrophil engraftment in the DARA-cohort, identified an interesting association of delayed platelet recovery with micromolecular MM subtype and, most significantly, showed that such post-transplant events did not limit engraftment success or increase mortality risk in the long term (9 months). The real-life setting is a point of strength of the study and the cohorts have been appropriately selected but a number of clinical and laboratory information about the two cohorts have not been provided limiting its value and explanatory potentials.

Therefore, before considering this work for publication major issues should be addressed by the authors:

1. Provide a protocol number given by the local institutional board;

2. Provide detailed data about pre-treatment and pre-ASC infusions baseline hematological and immunological laboratory markers (e.g. blood counts, IgG levels etc, complement levels, kidney function, etc.), also to be included in statistical analysis;

3. Provide detailed information about vaccination schedules of included patients both before and after ASCT;

4. Provide data about microbiological investigations and clinical-radiological features in febrile neutropenic patients;

5. Genetic analysis results should be described and statistically analyzed;

6. Report all the multivariables under the statistical analysis section, including the number of reiunfused CD34;

7. It would add significant weight to your study if you included the data pertaining to progression free survival after transplantation, incidence of hypogammaglobulinemia, infections, monoclonal component dynamics in the two cohorts.

Author Response

Dear Editor,

Thank you for your attention to our work. We appreciate the comments of the reviewers very much. We have tried to respond to all their comments, and certainly through their evaluation we have improved the quality of the work.

I hope it can now be suitable for publication.

REFEREE  1

  1. Provide a protocol number given by the local institutional board

The referee is absolutely right when saying that this point needs to be clarified. Thus, we are grateful for this comment because it allows us to elucidate this important aspect of our work. We added the sentence in methodological section (yellow) “This study was conducted as an observational analysis of drug effectiveness and safety, and in accordance with applicable guidelines, ethical committee approval was not deemed necessary. However, informed consent was obtained from all participants to ensure ethical standards were maintained.”

  1. Provide detailed data about pre-treatment and pre-ASC infusions baseline hematological and immunological laboratory markers (e.g. blood counts, IgG levels etc, complement levels, kidney function, etc.), also to be included in statistical analysis

To comply with the referee, we included in Table 1 baseline characteristics by treatment, data of WBC and PLT. Patients with abnormal renal function (serum creatinine value) >  2.0 mg/dL were excluded by the study,

  1. Provide detailed information about vaccination schedules of included patients both before and after ASCT

Thank you for this comment. In the discussion we wrote (highligheted in green) “Although not a goal of this study, we evaluated infectious events that occurred after engraftment for neutrophils and within six months post-transplantation, showing no difference in incidence between the two study groups. We follow the current indications of the European Society for Blood and Marrow Transplantation (EBMT), which recommend revaccination starting between 6 and 12 months after transplant [reference 38]. To reduce the risk of SARS-COV-2 infection, the primary immunization schedule was of 3 vaccine doses, starting from 3 to 6 months after transplant, followed by a booster dose after 3–4 months from the primary vaccine schedule”.

  1. Provide data about microbiological investigations and clinical-radiological features in febrile neutropenic patients;

We thank the reviewer for this much appreciated comment. We have detailed how we monitored and treated FN in the endpoints, with additional text highlighted in gray. “In the absence of clinically or microbiologically documented infection, empirical antibiotic was discontinued after 72 h of apyrexia and clinical recovery, irrespective of the neutrophils count.If no signs or symptoms of clinical deterioration were present, slow response to antibiotic treatment has been considered, particularly if accompanied by improvement in inflammatory markers such as C-reactive protein or procalcitonin.  If clinical conditions deteriorate, management steps were aggressive diagnostic workup (repeated blood cultures, additional testing for viruses and fungi, CT scan, BAL lavage in case of pneumonia, lumbar puncture in case of CNS symptoms)”

  1. Genetic analysis results should be described and statistically analyzed

Thank you for this comment. “We did not consider data related to patients' cytogenetic risk in the statistical analysis. With more mature follow-up, it will be interesting to consider data in terms of PFS and OS, stratifying patients according to cytogenetic features at disease onset”. We highlighted this weakness of the study, in yellow, in the discussion

  1. Report all the multivariables under the statistical analysis section, including the number of reinfused CD34

To comply with the request of the referee, we included CD34+>4 vs <4 into table 3b and found that this variable did not correlate with the outcome variable (HR: 1.17; 95% CI: 0.8-1.71, p value 0.42). We also amended a typo in table 3a univariable Cox analysis. The correct value of the effect of CD34+ on the outcome variable is now the following one: HR 1.396, 95% CI 0.9717-2.005, p value 0.071.

As for multivariable logistic regression on FN and diarrhea, after the inclusion of CD34+ into the model, the new results are reported below. Given the fact that the results did not materially change after the inclusion of this variable into the two models, we would prefer not including these new data into the paper, to avoid making the same paper heavier to read.

Multivariable logistic model on FN

Sign.

OR

95% C.I.per EXP(B)

Inferiore

Superiore

Gender M vs F

.609

1.198

.599

2.397

Age at transplant

.374

1.021

.976

1.067

D-VTd versus VTd

.026

2.217

1.100

4.467

CD34+ > 4 vs <4

.867

.937

.439

1.998

Multivariable logistic model on diarrhea

Sign.

OR

95% C.I.per EXP(B)

Inferiore

Superiore

Gender M vs F

.371

.634

.234

1.720

Age at transplant

.404

1.029

.962

1.101

D-VTd versus VTd

<.001

8.317

2.568

26.939

CD34+ > 4 vs <4

.422

.661

.241

1.816

  1. It would add significant weight to your study if you included the data pertaining to progression free survival after transplantation, incidence of hypogammaglobulinemia, infections, monoclonal component dynamics in the two cohorts.

We added in the discussion section (yellow) “Our study was not aimed at assessing survival, so we did not consider data related to patients' cytogenetic risk in the statistical analysis”. Our study was not aimed to analyse incidence of hypogammaglobulinemia, infections, monoclonal component dynamics in the two cohorts. Anyway, we added this aspect in discussion section “Although not a goal of this study, we evaluated infectious events that occurred after engraftment for neutrophils and within six months post-transplantation, showing no difference in incidence between the two study groups. We follow the current indications of the European Society for Blood and Marrow Transplantation (EBMT), which recommend revaccination starting between 6 and 12 months after transplant [38]. To reduce the risk of SARS-COV-2 infection, the primary immunization schedule was of 3 vaccine doses, starting from 3 to 6 months after transplant, followed by a booster dose after 3–4 months from the primary vaccine schedule” (green)

Reviewer 2 Report

Comments and Suggestions for Authors

The authors report first-line use of daratumumab in patients with multiple myeloma shows delayed neutrophil and platelet engraftment after autologous stem cell transplantation.

1.     The outline of the study is not clear. The authors should provide the study design in Figure.

2.     There is no patient information such as, karyotype, oncogenic mutation and treatment schedule in this study. The authors should clarify these things.

3.     Table 1 divides the data into two groups, but the proportions are not constant, so there is a bias.

4.     The prognostic data are not presented in this study.

5.     There is no description of side effects.

Author Response

Dear Editor,

Thank you for your attention to our work. We appreciate the comments of the reviewers very much. We have tried to respond to all their comments, and certainly through their evaluation we have improved the quality of the work.

I hope it can now be suitable for publication.

  1. The outline of the study is not clear. The authors should provide the study design in Figure.

To comply with the request of the referee, we added the protocol details in the Materials and Methods section (highlighted in pink) “After diagnosis, 60 consecutive NDMM patients were treated with four cycles of D-VTd according to the CASSIOPEIA trial schedule [6,7] before stem cell mobilization and transplantation. All patients received up to four 28-day, pre-transplant induction cycles of: V (1·3 mg/m² twice per week in week 1 [days 1 and 4] and week 2 [days 8 and 11] of each cycle) administrated by subcutaneous bolus injection into the thigh or abdomen; oral T (100 mg daily in all cycles); and oral or intravenous d (40 mg on days 1, 2, 8, 9, 15, 16, 22, and 23 of induction cycles 1 and 2 and days 1 and 2 of induction cycles 3 and 4 and 20 mg on days 8, 9, 15, and 16 of induction cycles 3 and 4). D was administered at a dose of 1800 mg SC bolus once weekly in induction cycles 1 and 2 and once every 2 weeks during induction cycles 3 and 4. DARA was injected into the subcutaneous tissue of the abdomen approximately 7.5 cm to the right or left of the navel over approximately 3-5 minutes. The study population has been compared with a historical group of 80 NDMM treated with VTd as induction therapy, treated between February 2021 to December 2023 without the use of DARA. [6,7] from February 2021 to December 2023. High-dose cyclophosphamide (2 g/m2) plus G-CSF were administered to mobilize peripheral blood stem cells in both arms. ”

  1. There is no patient information such as, karyotype, oncogenic mutation and treatment schedule in this study. The authors should clarify these things.

Thank you for this comment. Our study was not aimed at assessing survival, so we did not consider data related to patients' cytogenetic risk in the statistical analysis. With more mature follow-up, it will be interesting to consider data in terms of PFS and OS, stratifying patients according to cytogenetic features at disease onset. We highlighted this weakness of the study, in yellow, in the discussion.

  1. Table 1 divides the data into two groups, but the proportions are not constant, so there is a bias.

We thank the referee for this comment because it allows us to better clarify an aspect of our study. Although the proportions between the two groups are not constant, they are far to be statistically significant. The only variable that significantly differed between the two groups was CD34+ infused, a potential confounder specifically considered by us in both univariable and multivariable models.

  1. The prognostic data are not presented in this study.

We agree with the referee that the inclusion of prognostic data would have been an added value of our study. However, these data will be the object of a future study of our group in a dedicated paper.

  1. There is no description of side effects.

We than the referee for the comment. We added the sentence “No patients experienced unexpected side effects during transplantation. In particular, no cardiac, renal or hepatic toxicities were reported”, in Section 3.4 (highlighted in green).

Round 2

Reviewer 2 Report

Comments and Suggestions for Authors

none